# Clonal Hematopoiesis Mutations Are Present in Atherosclerotic Lesions in Peripheral Artery Disease

**DOI:** 10.3390/ijms24043962

**Published:** 2023-02-16

**Authors:** Petra Büttner, Julia Böttner, Knut Krohn, Ronny Baber, Uwe Platzbecker, Michael Cross, Steffen Desch, Holger Thiele, Sabine Steiner, Dierk Scheinert, Klaus H. Metzeler, Daniela Branzan

**Affiliations:** 1Department of Cardiology, Heart Center Leipzig at University of Leipzig, University of Leipzig, Strümpellstr. 39, 04289 Leipzig, Germany; 2Core Unit DNA-Technologies, Medical Faculty, University of Leipzig, Philipp-Rosenthal-Str. 55, 04103 Leipzig, Germany; 3Institute of Laboratory Medicine, Clinical Chemistry and Molecular Diagnostics, University of Leipzig, Paul-List-Strassse 13-15, 04103 Leipzig, Germany; 4Leipzig Medical Biobank, University of Leipzig, Strümpellstr. 39, 04289 Leipzig, Germany; 5Department of Hematology, Cellular Therapy and Hemostaseology, Leipzig University Hospital, Liebigstrasse 20, 04103 Leipzig, Germany; 6Division of Angiology, Department of Internal Medicine, Neurology and Dermatology, Leipzig University Hospital, Liebigstrasse 20, 04103 Leipzig, Germany; 7Helmholtz Institute for Metabolic, Obesity and Vascular Research (HI-MAG) of the Helmholtz Zentrum Munich at the University of Leipzig and University Hospital Leipzig, Rosenthal-Straße 27, 04103 Leipzig, Germany; 8Visceral, Transplantation, Thorax and Vascular Surgery, Leipzig University Hospital, Liebigstrasse 20, 04103 Leipzig, Germany

**Keywords:** clonal hematopoiesis, peripheral artery disease, atherosclerosis, *TET2*, *DNMT3A*

## Abstract

Clonal hematopoiesis (CH)-associated mutations increase the risk of atherosclerotic cardiovascular diseases. However, it is unclear whether the mutations detected in circulating blood cells can also be detected in tissues associated with atherosclerosis, where they could affect physiology locally. To address this, the presence of CH mutations in peripheral blood, atherosclerotic lesions and associated tissues was assessed in a pilot study of 31 consecutive patients with peripheral vascular disease (PAD) who underwent open surgical procedures. Next-generation sequencing was used to screen the most commonly mutated loci (*DNMT3A, TET2, ASXL1* and *JAK2*). Twenty CH mutations were detected in peripheral blood of 14 (45%) patients, 5 of whom had more than one mutation. *TET2* (11 mutations, 55%) and *DNMT3A* (8 mutations, 40%) were the most frequently affected genes. Altogether, 88% of the mutations detectable in peripheral blood were also present in the atherosclerotic lesions. Twelve patients also had mutations in perivascular fat or subcutaneous tissue. The presence of CH mutations in PAD-associated tissues as well as in blood suggests that CH mutations may make a previously unknown contribution to PAD disease biology.

## 1. Introduction

Atherosclerotic diseases are responsible for the most deaths in Western civilizations and inflammation is one of the main features in the progression of atherosclerosis [1]. Recent research has revealed a link between inflammatory processes and clonal hematopoiesis, characterized by somatic mutations in one of a defined set of genes in the hematopoietic stem cell pool [2,3]. When these mutations bestow a competitive advantage in the stem and progenitor cell population, clonal expansion results in the affected variant becoming detectable as a disproportionately large fraction of the circulating immune cell progeny, where they may influence inflammatory properties [2]. Age-associated clonal hematopoiesis (CH) has not only been associated with a higher risk for myelodysplasia and leukemia, which arise from CH at a rate of around 1% per year [3,4], but also with cardiovascular adverse events [4,5]. Several studies have shown correlations between CH and coronary artery disease [3], chronic ischemic heart failure [6], degenerative aortic valve stenosis [7] and effects on cardiovascular as well as general mortality [3,5,6]. While the evidence for an association between CH and atherosclerosis is now strong, the pathomechanisms in humans are not well understood. In particular, the presence and functional relevance of CH mutations in human atherosclerotic plaques have yet to be elucidated.

A major obstacle to further research is the limited access to human samples, especially in coronary artery disease. However, atherosclerosis is also the primary cause of peripheral artery disease (PAD) [8], causing a narrowing of the arteries supplying the lower limbs. Patients with PAD usually present for treatment because of intermittent chronic limb-threatening ischemia (CLTI). The anatomical location and extent of the arterial lesions causing symptoms determine the revascularization options [8]. Several studies have shown the efficacy of open surgical as well as interventional revascularization of atherosclerotic stenotic and occluded peripheral arteries in terms of symptom relief and quality of life in PAD patients [9]. In addition, the surgical revascularization technique offers the opportunity to collect not only blood but also atherosclerotic material and associated tissue such as perivascular fat, collateral vessels and subcutaneous tissue. The aim of this study was therefore to assess the presence of CH mutations in peripheral blood, atherosclerotic plaques and associated tissues in patients with PAD as a model of atherosclerotic disease.

## 2. Results

### 2.1. Patients and Tissue Sampling

This pilot study included 31 PAD patients. Patient demographics and clinical characteristics are presented in Table 1. Patients had a median age of 68 (IQR 60–84) years and the majority (58%) were treated for CLTI. In 25 (81%) patients, the treated arterial lesion was classified as stenosis based on preoperative imaging with computed tomography angiography, magnetic resonance angiography, or digital subtraction angiography as recommended by the guidelines [8]. Six (19%) patients underwent revision surgical procedures of a previous open arterial reconstruction. Whole blood, atherosclerotic lesions and subcutaneous tissue were obtained from the entire cohort. Collateral vessels and perivascular fat were intraoperatively collected from 29 and 30 patients, respectively. Since two patients died of acute myocardial infarction in hospital before their follow-up visit and two patients were lost to follow-up, no blood cell enrichments and CH analyses of blood sub-populations from these four patients were performed.

### 2.2. CH Detection in PAD Patients

CH mutations were detected in 14 PAD patients (45%), 5 (16%) of whom had at least two mutations (Table 2). The demographics, clinical characteristics, and laboratory parameters of PAD patients with and without CH were similar (Table 1). CLTI was the indication for treatment in 18 (58%) patients, and CHIP was detected in 9 (50%) of these. CHIP was also detected in 5 (38%) of 13 patients treated for claudication. By trend, occlusion of the atherosclerotic artery was more frequently encountered in patients with CH than in those without (*p* = 0.067).

A total of 20 different CH mutations were detected in the entire cohort. By analyzing blood sub-populations, atherosclerotic lesions and other tissues in addition to peripheral blood, the CH mutation detection rate in PAD patients increased from 45% to 70%. The most frequently affected gene was *TET2* (tet methylcytosine dioxygenase 2) with 11 mutations (55%). The second most affected gene was *DNMT3A* (DNA [cytosine-5]-methyltransferase 3A) with 8 mutations (40%). There was only one mutation detected in *ASXL1* (additional sex combs-like 1, transcriptional regulator 1).

### 2.3. CH Detection in Whole Blood and Blood Sub-Populations

CH mutations were found by automated calling in 13 (42%) patients. Interestingly, CH was not detected in peripheral blood by automated calling in patient 15, but was found in the purified monocyte fraction. Manual re-evaluation (see Appendix A) revealed the same CH mutation to be present both in peripheral blood and in the other cell populations at a low frequency. Overall, all 20 CH mutations were found in peripheral blood or at least in one of the analyzed blood sub-populations. Eight mutations had their highest detected VAF specifically in monocytes (*n* = 4), progenitors (*n* = 4) or residual cells (*n* = 2).

### 2.4. CH Detection in Atherosclerotic Lesions

CH mutations were detected in atherosclerotic lesions from 13 patients (42% of the PAD cohort, 93% of CH-positive PAD patients). A total of 15 (75%) of the 20 CH mutations detected were also present in arterial atherosclerotic lesions. The VAF in arterial lesions was similar to or lower than that in peripheral blood, except for three mutations where VAF was highest in arterial lesions (Table 2).

### 2.5. CH Detection in Arterial Collaterals, Perivascular Fat and Subcutaneous Tissue

CH mutations were identified by automated calling in collateral arterial samples from 9 of the 14 CH-positive PAD patients. CH mutations were also detected in perivascular fat and/or subcutaneous tissue from all 14 patients and, while VAF was low in 8 patients, the remaining 6 had CH in these tissues at a VAF above 0.015. One patient had a mutation in the collateral vessel, but not in the atherosclerotic lesion.

## 3. Discussion

In 2014, Jaiswal et al. were the first to describe an association between atherosclerotic events and the presence of CHIP with a hazard ratio of 2.0 (95% confidence interval (CI) 1.1–1.8) for coronary artery disease (CAD) [4]. To the best of our knowledge, this pilot study demonstrates for the first time that CH mutations are a common finding in PAD patients not only in peripheral blood but also in atherosclerotic lesions, arterial collaterals, perivascular fat and subcutaneous tissue. By enrichment of CD14^+^ monocytes and CD34^+^ progenitor cells from the peripheral blood, we demonstrated that both populations harbored CH mutations, as did the residual CD14^−^ CD34^−^ population.

In this study, 15 CH mutations were detected by automated calling in the peripheral blood of 13 PAD patients (42%), 9 of whom had CLTI. The extended screening approach including blood sub-populations, atherosclerotic lesions and other tissues led to the identification of CH in one additional patient, increasing the overall CH detection rate to 45%. Furthermore, by analyzing blood cell sub-populations and tissue samples in addition to whole peripheral blood, the number of detected variants increased by 25% to 20 mutations in total. Earlier studies in patients with other diseases using less sensitive approaches identified a CH prevalence of 6% in 60–69 year old subjects, and 12% in those aged ≥ 70 years [4,14,15], while a highly sensitive approach recently reported a 50% prevalence [16]. Importantly, the sensitivity of CH mutation detection depends both on the sequencing method and on the choice of a targeted (candidate) vs. non-targeted (genome-wide) strategy. A direct comparison therefore presupposes the same analysis strategy. Using the same method as we used for this current study, we recently detected CH mutations in 29% of 446 patients with cardiogenic shock (CS) after acute myocardial infarction [17]. The median age of 69 years in the CS cohort was comparable to that in the PAD cohort. The balance of mutations differed between CS and PAD, with *DNMT3A* being the most frequently mutated gene (47%) in CS followed by *TET2* (36%), while in the PAD cohort, the most common target of mutation was *TET2* (55%) followed by *DNMT3A* (40%). This is consistent with experimental studies showing a specific mechanistic connection between mutated *TET2* and atherosclerosis [18].

*TET2* catalyzes the oxidation of 5-methylcytosine, an epigenetic process influencing transcription regulation [2]. Mutations in *TET2* typically inactivate or diminish the enzyme’s function as a negative regulator of pro-inflammatory macrophage activation and cytokine secretion [19]. In a murine heart failure model, *Tet2* deficiency negatively affected cardiac remodeling and increased pro-inflammatory interleukin (IL)-1β [19]. In another study, mice with heterozygous *Tet2* knockout developed larger atherosclerotic lesions and their macrophages showed increased expression of genes encoding cytokines, including IL-1β [3,18]. In vitro, macrophages with *TET2*-repressing mutations expressed the pro-inflammatory IL-1β, IL-6 and inflammasome NLRP3 in a pattern typically associated with macrophage activation in infectious diseases [20].

*DNMT3A* was the second most commonly mutated gene in PAD patients. *DNMT3A* plays a role in the suppression of the type I interferon response and the orchestration of an appropriate immune response to infections [20]. *DNMT3A* mutated monocytes from patients with heart failure have been shown to express higher mRNA levels of IL-1β, IL-6 and IL-8, macrophage inflammatory chemokines CCL3 and CCL4 and the inflammasome NLRP3. In addition, gene expression of molecules associated with T-cell stimulation and monocyte–T-cell interaction was increased in these patients [21].

In this study, enriched CD14^+^ monocytes and CD34^+^ hematopoietic progenitors were analyzed separately for the presence of CH mutations. It should be noted that we are unable from our results to extrapolate from population-specific VAFs to the expected contribution of each population to the VAFs seen in unseparated peripheral blood, firstly because of the uncertainties and variation in the efficiencies and yields of the MACS purifications, and secondly because we have no standardized FACS data of individual peripheral blood samples. The data from enriched populations should therefore be regarded as semi-quantitative but they suffice nonetheless to reveal differences between PAD cases in the distribution of CH mutations between blood sub-populations.

CD34 is a cell surface glycoprotein involved in cell–cell adhesion and is considered a widely expressed marker in circulating progenitor cells. Indeed, we found CH mutations in the CD34^+^-enriched cell fraction of eight PAD patients. Intriguingly, four of them, all *TET2* mutations, had the highest VAF in this cell compartment. CD14^+^ monocytes were also enriched for specific analysis, as these are in the focus of CH-associated pro-atherogenic pathomechanisms [18,20]. In fact, CH mutations were found in CD14^+^ enrichments of nine patients with a VAF up to 0.26 in one patient. In some cases, CH mutations were present at high frequencies in residual cells that constitute a mix of granulocytes, T- and B-Lymphocytes and natural killer cells. It has been reported that *TET2* and *DNMT3A* mutations occur in T-lymphocytes and alter their function and expression profiles [20]. The effects of CH mutations on B-lymphocytes and natural killer cells remain largely uncharacterized, but maturation and function might also be influenced [20].

In this cohort pilot study, 75% of atherosclerotic lesions from CH carriers were positive for at least one CH mutation. Next to the native resident smooth muscle cells and endothelial cells [1], macrophages are the most abundant leukocyte type in atherosclerotic vessels, present at all progression stages [2]. To a lesser extent, polymorphonuclear cells are also involved in certain atherosclerotic processes [2]. Importantly, both cell types are of myeloid origin and have been reported to harbor CH mutations [20]. We are unable to distinguish which cells have contributed CH mutations into the atherosclerotic lesion, especially since there was a diverse pattern of blood sub-populations harboring CH mutations, as outlined above. However, because macrophages have been frequently reported to harbor CH mutations and their role in atherosclerosis is preeminent, we consider these to be the most likely source. Nevertheless, neutrophil granulocytes have also been associated with CH in vasculitis. It has been hypothesized that CH mutations may interfere with the formation and maturation of neutrophilic granules involved in inflammatory and cytotoxic processes [22]. As this study detected CH mutations also in the residual cells from cell enrichments, we cannot exclude a contribution of polymorphonuclear cells or lymphocytes to the CH signal detected from atherosclerotic lesions.

Surprisingly, CH mutations were also detected in arterial collaterals that developed next to the occluded or stenotic artery. These vessels are thought to evolve de novo by arteriogenesis or by enlargement of pre-existing arteriolar connections involving cell proliferation and tissue remodeling [23]. On preoperative imaging, none of the patients with CH-positive collateral arteries showed signs of atherosclerotic collateral calcification. However, early atherosclerotic pathomechanisms involving CH-positive cells may have already been initiated. Collateral formation is triggered by shear stress, leading to the secretion of monocyte chemoattractant protein MCP1 by smooth muscle cells, possibly resulting in monocyte invasion into the newly formed vessel [23]. Finally, circulating endothelial progenitor cells (EPC) may contribute to repair processes in atherosclerotic arteries as well as to collateral artery formation and may thus be important in the context of CH. It has been described that PAD patients have impaired EPC count, function and proliferation indices. In addition, the severity of PAD, e.g., lower ankle–brachial index and higher intima–media thickness, correlates with inferior EPC properties [24]. Although this analysis detected CH mutations in circulating CD34^+^ cells, we are aware that only a small fraction of CD34^+^ cells are EPCs. Therefore, these results cannot be generalized to EPCs, and further studies are needed to determine whether CH-positive EPCs contribute to atherosclerotic processes.

Most CH-positive PAD patients presented CH mutations in subcutaneous tissue or perivascular fat. Histological characterization of these tissues was not performed, but they represent mostly adipocytes, as previously described [25]. CH mutations in human adipose tissue have not been described previously. It may be hypothesized that infiltrating immune cells, most likely macrophages [26], have introduced CH mutations into these tissues. Recently, *Tet2* loss-of-function mutations have been shown in mice to increase IL-1β secretion from adipose tissue and aggravate age- and obesity-related insulin resistance [27]. Perivascular fat distribution [25] and adipokine signaling [28] influence the progression of PAD and adipose tissue-derived retinol binding protein 4 predicts major adverse events in PAD patients [28]. Thus, further studies should analyze whether and how adipose tissue signaling is affected by CH.

PAD is a multifactorial disease in which lifestyle factors such as smoking, advanced age and common diseases such as hypertension, hyperlipidemia and diabetes mellitus contribute to the manifestation and progression of the disease. This study introduces CH mutations as an additional relevant risk factor. Importantly, smoking [2], advanced age [2] and diabetes mellitus [27] are also independently associated with the occurrence of CH. Obesity is associated with sterile inflammation leading to a chronic low-level inflammatory status. A pro-inflammatory environment is also observed post-infection or after TNF-α administration in animal models, and presumably favors the proliferation of *TET2*-mutant stem cells, triggering a higher proportion of circulating *TET2*-mutant immune cells [20]. Thus, classical PAD risk factors may favor the establishment of the novel risk factor CH, while CH in turn favors the establishment of common risk factors such as diabetes mellitus [5]. This interplay between common risk factors and CH is consistent with the observation that individuals who have an optimal risk factor profile in middle age are very unlikely to develop coronary artery disease at older ages [29].

The current results may help to develop novel pharmaceutical interventions in PAD, potentially tailored to individual CH mutations. Interestingly, irrespective of CH, IL-1β inhibitors have already been found to improve cardiovascular outcomes in patients with stable coronary artery disease [30]. Similarly, IL-1β inhibitors such as Anakinra, Rilonacept, Gevokizumab or Canakinumab [31] and selective NLRP3 inflammasome inhibitors [32] may be useful for PAD patients with *TET2* loss-of-function mutations [2,20,27].

CH is a promising area for future mitigation of cardiovascular risk and hematologic malignancies. Given the broad clinical implications of the entity of CH, there is a clear need for surveillance and potential preventive measures. Given the reciprocal links between CH, malignancies and CVD, closely intertwined interdisciplinary communication is likely essential for effective clinical management [33,34].

### Limitations

The results reported here are based on a relatively small cohort, precluding multivariable analyses, and the study needs to be replicated in larger cohorts. The tissues used for sequencing were not examined histologically, so neither the general cell composition nor the degree of infiltration by immune cells are known.

The blood sub-populations analyzed were enrichments rather than highly purified populations. It cannot be excluded that contaminating cell populations contributed to the mutation signals. Nevertheless, VAF was mostly significantly increased in the specific cell populations compared with whole blood, correspondingly to the enrichment factor. We consider this as strong evidence that the enriched cells are the source of CH mutations.

All tissue samples were extensively rinsed and were macroscopically free from any blood traces before DNA was isolated, even so, it cannot be excluded that blood cells were still present in these tissues.

## 4. Methods and Materials

### 4.1. Patients

#### 4.1.1. Study Cohort

Consecutive patients with symptomatic chronic atherosclerotic PAD (*n* = 43) scheduled for open arterial revascularization of the lower limbs at University Hospital Leipzig between February 2021 and June 2021 were prospectively screened. PAD was diagnosed based on symptoms of intermittent claudication, rest pain and/or skin lesions, ankle–brachial index < 0.80, peripheral artery stenosis > 50% documented by duplex ultrasound and computed tomography angiography, magnetic resonance angiography or digital subtraction angiography as recommended by the guidelines [8]. Two patients refused to participate and 10 patients were excluded due to logistical reasons, the study being paused for one month. The final cohort consisted of 31 patients. The study was approved by the local Ethical Committee (Medical Faculty, University Leipzig, registration number 598/20-ek) and all 31 patients gave written informed consent in accordance with the Declaration of Helsinki.

#### 4.1.2. Surgical Procedure and Follow-Up Visits

Baseline clinical data, medication (antiplatelet therapy, oral anticoagulation and statins), imaging findings and laboratory data were collected prospectively. Patients were defined as claudicants or CLTI according to the ESVS guidelines [8]. Computed tomography angiography or digital subtraction angiography were performed preoperatively in all patients to localize arterial lesions and plan open revascularization procedures; lesions were classified as stenosis or occlusion. The degree of lesion calcification was calculated using the PACSS score [35]. Grade 4 lesions were considered as severe calcifications and lesions below grade 4 as moderate calcifications. After induction of anesthesia for open revascularization of the lower limbs, peripheral blood samples were collected. After the skin incision on the marked operated side, subcutaneous fat and perivascular tissue were collected. When arterial collaterals were removed to access the atherosclerotic arteries, they were also harvested. After intravenous administration of 5000 IU unfractionated heparin, arterial cross-clamping followed by arteriotomy and removal of the atherosclerotic plaque was performed, and plaque samples were collected. The arterial flow to the foot was reconstructed according to guidelines [8]. In case of unsuccessful revascularization, major amputation of the lower limb was performed, the atherosclerotic plaques being collected from the transected artery. Peripheral blood and intraoperative tissue samples were immediately transported to the research laboratory at Heart Center Leipzig at the University of Leipzig for processing and to the Institute for Pathology Leipzig for long-term storage in the Leipzig Medical Biobank. Follow-up examinations were scheduled one month after surgery and included clinical and ultrasound examination and laboratory tests.

### 4.2. Sample Processing and Biobanking

Tissue samples collected during surgery were instantly placed in sterile 1 mL cryotubes for snap freezing (Greiner Bio-One, Frickenhausen, Germany), or in sterile MACS tissue storage solution (Miltenyi Biotech, Bergisch-Gladbach, Germany) for DNA isolation. A fraction of at least a 2 mm × 2 mm × 2 mm tissue sample was washed in phosphate-buffered saline and used for DNA isolation. Peripheral blood was collected in S-Monovettes (Sarstedt, Nürmbrecht, Germany) containing EDTA as an anticoagulant. All samples for direct DNA extraction were processed within 18 h. If samples had to be stored overnight, they were kept at 4 °C.

Enrichment of blood sub-populations was performed from 5 mL peripheral blood collected at the follow-up visit. CD14^+^ monocytes were isolated (StraightFrom^®^ Whole Blood CD14 MicroBeads (Miltenyi Biotech)) according to the manufacturer’s recommendation. CD34^+^ progenitor cells (MACSprep™ Chimerism CD34 MicroBead Kit) were enriched with minor modifications of the recommended protocol: the remaining cells from CD14^+^ enrichment were used as a source from which to isolate CD34^+^ cells, having the advantage that monocytes were already depleted. All remaining cell populations, hereafter termed residual cells, were also collected and analyzed. CD14^+^ and CD34^+^ enrichments were checked for purity using flow cytometry (FACS Calibur, Becton Dickinson, Franklin Lakes, NJ, USA). CD34^+^ enrichments and the residual cells were free from CD14^+^ cells (detection rate < 1.5%).

DNA was isolated using the DNeasy Blood and Tissue kit (Qiagen, Hilden, Germany). Proteinase K digestion for the tissue samples was extended up to one hour and lysates were additionally passed through a QIAshredder column (Qiagen).

### 4.3. Sequencing

For CH mutation analysis, we developed a custom amplicon-based gene panel (targeting 25 kb of genomic DNA sequence), focusing on detection of the most frequent hematopoietic and lymphoid-attributed mutations in the Catalogue of Somatic Mutations in Cancer (COSMIC) database in the four most frequently mutated driver genes in CH: *DNMT3A, TET2, ASXL1* and *JAK2*. Methods were recently described [17]. In brief, a total of 50 ng DNA was PCR-amplified in four multiplex reactions primed with oligonucleotide sequences of the genes detailed above.

DNA libraries were prepared from 10 ng of the pooled PCR products using the Illumina Nextera XT kit according to the manufacturers recommendations (Illumina, San Diego, CA, USA). Up to 120 libraries were used for sequencing at a total concentration of 2 nM. Sequencing of 2 × 150 bp was performed with an Illumina NextSeq 550 sequencer at the sequencing core facility of the Faculty of Medicine (University of Leipzig). Demultiplexing of raw reads, adapter trimming and quality filtering were carried out according to Stokowy et al. [36]. After mapping against the human reference genome (hg38) using bwa [37] (version 0.7.17-r1188), we employed the VarDict algorithm for variant calling [38] and ANNOVAR for variant annotation [39]. If a mutation was detected in at least one patient sample by VarDict, all other samples were also screened manually. Detailed information on the screening approach can be found in the supplemental methods, Appendix A. Only exonic or splice site mutations were included. Variants occurring in ≥ 5% of all studied patients were considered probable technical artefacts and excluded. Variants that were covered by fewer than 200 reads were also filtered out, as were synonymous SNVs and mutations known to have a minor allele frequency of 0.002 or more in the gnomAD database. To exclude rare germline variants, mutations with VAF > 0.35 were excluded unless they were stopgain, frameshift or splice-site mutations.

### 4.4. Statistical Analyses

Data were analyzed using SPSS version 28.0 (IBM, Armonk, NY, USA). Categorical variables are presented as number and percentages, and continuous variables as median values with interquartile range (IQR). Normal distribution of continuous data was checked using Kolmogorov–Smirnov and Shapiro–Wilk tests. Statistical differences between patients with and without CH were detected using the Mann–Whitney U-test for continuous data and the Chi-square test for independence for categorical data. P < 0.05 was considered statistically significant.

## Figures and Tables

**Table 1 ijms-24-03962-t001:** Characteristics of patients with peripheral artery disease (PAD). The *p*-values describe the comparison of patients with CH mutations and without CH mutations and were calculated using Chi-square test (categorical variables, presented as number and percentage) or Mann–Whitney U-test (numerical variables, presented as median and inter quartile range). CLTI—critical limb-threatening ischemia, COPD—chronic obstructive pulmonary disease, CRP—C-reactive peptide, HbA1c—glycohemoglobin, GFR—glomerular filtration rate, PACSS—peripheral arterial calcium scoring system, BP—blood pressure.

	Total 31 (100%)	CH 14 (45%)	No CH 17 (55%)	*p*-Value
Male	26 (84%)	11 (79%)	15 (88%)	0.636
Age (years)	68 (60–84)	68 (57–88)	68 (61–84)	0.953
Hypertension	28 (90%)	14 (100%)	14 (82%)	0.232
BP systolic (mmHg)	144 (123–193)	141 (119–191)	149 (122–198)	0.891
BP diastolic (mmHg)	74 (57–89)	71 (54–92)	75 (61–88)	0.336
Diabetes mellitus	14 (45%)	6 (43%)	8 (47%)	1.000
Glucose (mmol/mL)	5.9 (4.7–11.4)	5.9 (4.9–19.1)	5.9 (4.6–9.9)	0.739
HbA1c (%)	42.2 (34.7–74.8)	42.7 (33.5–82.5)	41.3 (34.9–65.9)	0.720
HbA1c (mmol/mL)	7.7 (5.2–9.3)	7.9 (4.6–9.3)	7.7 (5.1–9.5)	0.830
Coronary artery disease	10 (32%)	5 (36%)	5 (29%)	1.000
Smoker	21 (68%)	11 (79%)	10 (59%)	0.280
COPD	6 (19%)	1 (7%)	5 (29%)	0.185
Obesity (BMI > 30 kg/m^2^) [10]	6 (19%)	2 (14%)	4 (24%)	0.664
Body mass index (kg/m^2^)	24.6 (20.9–31.2)	25.5 (19.2–30.8)	24.6 (21.8–36.2)	0.769
Hyperlipoproteinemia [11]	25 (81%)	13 (93%)	12 (71%)	0.185
Total Cholesterol (mmol/L)	3.9 (3.0–5.6)	3.8 (2.8–6.9)	4 (2.9–5.3)	0.560
LDL-Cholesterol (mmol/L)	2.3 (1.5–4.0)	1.9 (1.2–5.3)	2.4 (1.6–3.3)	0.212
HDL-Cholesterol (mmol/L)	1.3 (0.8–2.1)	1.2 (0.8–2.2)	1.3 (0.9–2.0)	0.527
Triglycerides (mmol/L)	1.5 (0.8–2.7)	1.7 (0.8–3.3)	1.5 (0.9–2.5)	0.432
Anemia [12] (Hb < 11.8 g/dL)	11 (35%)	5 (36%)	6 (35%)	1.000
Hb (g/dL)	12.3 (8.4–14.8)	12.6 (7.4–14.8)	12.3 (8.2–15.3)	0.860
RDW %	13.9 (12.6–18.2)	13.8 (12.2–16.8)	14.4 (12.6–18.9)	0.544
Renal Insufficiency (GFR < 60) [13]	12 (39%)	3 (21%)	9 (53%)	0.138
Creatinine (µmol/L)	87 (60.6–142.2)	83.5 (56–144)	109 (69.2–146.4)	0.138
GFR (mL/min/1.73 m²)	66 (36–99.8)	72 (33.5–102.5)	59 (34–94.2)	0.100
CRP (mg/L)	7.3 (1.1–166.9)	6.2 (0.6–177.4)	7.4 (1.8–137.5)	0.739
Medication				
Antiplatelet therapy	26 (84%)	11 (79%)	15 (88%)	0.636
Oral anticoagulation	3 (10%)	2 (14%)	1 (6%)	0.576
Statins	22 (71%)	12 (86%)	10 (59%)	0.132
Clinical Presentation of PAD				
Claudication	13 (42%)	5 (36%)	8 (47%)	0.717
CLTI	18 (58%)	9 (64%)	9 (53%)	
Morphology of Arterial lesion				
Stenosis	25 (81%)	9 (64%)	16 (94%)	0.067
Occlusion	6 (19%)	5 (36%)	1 (6%)	
Severe calcification of lesion (PACSS)	18 (58%)	10 (71%)	8 (47%)	0.275
Re-do surgical procedure	6 (19%)	3 (21%)	3 (18%)	1.00

**Table 2 ijms-24-03962-t002:** CH driver mutations in patients with peripheral artery disease. First column—index patient (IP) in whom the variant was detected. Second/third column—affected gene and base position. Fourth–seventh column—variant allele frequencies (VAF) in blood. PB—peripheral blood; enriched blood sub-populations: CD14^+^—monocytes, CD34^+^—progenitor cells, residuals—PB depleted of monocytes and progenitor cells. Eighth–eleventh column—VAF in tissues, ST = subcutaneous tissue, PVT = perivascular tissue. Coloring indicates high (red), medium (pink) and low (white) VAF. Grey coloring indicates variant allele not found (0.00), VAF under detection threshold (marked with asterisk), l.c. = low coverage (fewer than 200 reads) due to low cell yield from enrichment, n.a. = not analyzed as sample was not accessible.

			Blood				Tissue			
IP	Gene	Base Position	PB	CD14^+^	CD34^+^	Residuals	Plaque	Collateral	ST	PVT
			*n* = 17	*n* = 11	*n* = 8	*n* = 10	*n* = 15	*n* = 10	*n* = 11	*n* = 11
1	*TET2*	105237044	0.0246	0.0967	0.1572	0.0558	0.0204	0.0058	0.0192	0.0188
		105235051	0.0000	0.0074	0.0000	0.0309	0.0000	0.0000	0.0023	0.0000
2	*TET2*	105259638	0.0395	0.1976	0.0476	0.1294	0.0000	0.0134	0.0214	0.0220
3	*DNMT3A*	25234307	0.0416	0.1452	0.0105	0.1009	0.0484	0.0092	0.0129	0.0153
4	*DNMT3A*	25240379	0.0653	0.0637	l.c.	0.0000	0.0130	0.0106	0.0305	0.0350
7	*DNMT3A*	25240699	0.0288	n.a.	n.a.	n.a.	0.0124	0.0032 *	0.0027 *	0.0037 *
8	*DNMT3A*	25246671	0.0267	0.0255	0.0000	0.0322	0.0053	0.0090	0.0096	0.0082
	*TET2*	105259678	0.0032 *	0.0167	0.1261	0.0016 *	0.0000	0.0000	0.0003 *	0.0010 *
9	ASXL1	32434825	0.0169	l.c.	l.c.	l.c.	0.0121	0.0044 *	0.0000	0.0029 *
	*TET2*	105275568	0.0004 *	0.0005 *	0.0603	l.c.	0.0010 *	0.0005 *	0.0000	0.0002 *
10	*DNMT3A*	25244175	0.0266	n.a.	n.a.	n.a.	0.0065 *	0.0045 *	0.0027 *	0.0124
	*TET2*	105236178	0.0527	n.a.	n.a.	n.a.	0.0395	0.0218	0.0129	0.0407
		105275086	0.0352	n.a.	n.a.	n.a.	0.0503	0.0442	0.0119	0.0301
15	*TET2*	105235270	0.0120	0.0221	0.0153	0.0051	0.0041	0.0029	0.0079	0.0019
19	*TET2*	105243673	0.2022	0.2648	0.2576	0.1973	0.0582	0.0916	0.0577	0.0465
21	*TET2*	105237312	0.0253	0.0159	0.0526	0.0168	0.0186	0.0014 *	0.0087 *	0.0020 *
25	*DNMT3A*	25234323	0.0423	n.a.	n.a.	n.a.	0.0037	0.0043	0.0063	0.0048
26	*TET2*	105235103	0.0192	n.a.	n.a.	n.a.	0.0175	0.0031 *	0.0047 *	0.0019 *
28	*DNMT3A*	25240699	0.0342	0.0181	l.c.	0.0203	0.0438	0.0026 *	0.0039 *	0.0029 *
		25246755	0.0132	0.0085 *	0.0080 *	0.0116	0.0271	0.0001 *	0.0010 *	0.0047 *

## Data Availability

Data are available on reasonable request.

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
