# Peer review of "Clonal Hematopoiesis Mutations Are Present in Atherosclerotic Lesions in Peripheral Artery Disease"

_ijms, 2023, doi:10.3390/ijms24043962_

Round 1
Reviewer 1 Report (New Reviewer)
The authors should be congratulated on writing this manuscript.
1. Table 1: Please expand and define the abbreviation "RR" in the legend.
2. Subheading 2.1. "Patients and tissue sampling": How was vascular stenosis/occlusion identified? Did patients have a CT scan with contrast, an invasive angiogram or both? This is important clinical information that guides revascularization strategy. This has been mentioned later in line 277, but please mention something simple like "stenosis by angiography" in subheading 2.1 to improve readability.
3. Line 279: Please mention CT angiography or invasive angiography.
4. Table 1: Authors mention re-do surgical procedure which is an important outcome and quality metric in these patients. Was there a difference in recurrent CLTI or incidence of amputation between the CH and No CH groups? If so, please include in table.
5. Please review entire manuscript for appropriateness and uniformity of the Oxford comma.
6. Although the focus of this article is the role of clonal hematopoiesis in PAD, please consider adding one or two brief paragraphs on the clinical correlation of this topic. This may increase readership, and outlining the clinical implications of such an important problem that affects millions of people worldwide will only add to the relevance of this paper.
Author Response
The authors should be congratulated on writing this manuscript.
Thank you for your evaluation and your time.
- Table 1: Please expand and define the abbreviation "RR" in the legend.
For a better understanding, we have replaced RR with BP, which stands for blood pressure. We have defined this abbreviation in line 84 of the Manuscript.
- Subheading 2.1. "Patients and tissue sampling": How was vascular stenosis/occlusion identified? Did patients have a CT scan with contrast, an invasive angiogram or both? This is important clinical information that guides revascularization strategy. This has been mentioned later in line 277, but please mention something simple like "stenosis by angiography" in subheading 2.1 to improve readability.
The vascular stenosis/occlusion was identified on computed tomography angiography, magnetic resonance angiography, or digital subtraction angiography based on the “Recommendations on imaging in patients with lower extremity artery disease” from the 2017 ESC Guidelines on the Diagnosis and Treatment of Peripheral Arterial Diseases, in collaboration with the European Society for Vascular Surgery (ESVS). We have improved the readability by adding this information in Subheading 2.1. We quote (line 68-71): “In 25 (81%) patients, the treated arterial lesion was classified as stenosis based on preoperative imaging with computed tomography angiography, magnetic resonance angiography, or digital subtraction angiography, as recommended by the guidelines.”
- Line 279: Please mention CT angiography or invasive angiography.
We have used computed tomography angiography, magnetic resonance angiography, or digital subtraction angiography to detect stenosis/occlusion as recommended by the guidelines (Reference 8). We have added this information in the manuscript in subheading 4.1.1 and we quote (line 285-289): “PAD was diagnosed based on symptoms of intermittent claudication, rest pain and/or skin lesions, ankle-brachial index (ABI <0.80), peripheral artery stenosis >50% documented by duplex ultrasound and computed tomography angiography, magnetic resonance angiography, or digital subtraction angiography as recommended by the guidelines.”
- Table 1: Authors mention re-do surgical procedure, which is an important outcome and quality metric in these patients. Was there a difference in recurrent CLTI or incidence of amputation between the CH and No CH groups? If so, please include in table.
From 31 patients, six patients underwent re-do surgical revascularization procedures, two due to severe claudication and four due to CLTI: Only three patients with Re-do procedures had CH mutations. Due to this small number a statistical analysis was not considered meaningful and was therefore not performed
- Please review entire manuscript for appropriateness and uniformity of the Oxford comma.
The entire manuscript was checked and corrected by a native speaker.
- Although the focus of this article is the role of clonal hematopoiesis in PAD, please consider adding one or two brief paragraphs on the clinical correlation of this topic. This may increase readership, and outlining the clinical implications of such an important problem that affects millions of people worldwide will only add to the relevance of this paper.
We have added a short paragraph on the clinical implication of CH and PAD in the discussion section (line 261-265) and quote: “CH is a promising area for future mitigation of cardiovascular risk and hematologic malignancies. Given the broad clinical implications of the entity CHIP, there is a clear need for surveillance and potential preventive measures. Given the reciprocal links between CHIP, malignancies, and CVD, closely intertwined interdisciplinary communication is likely essential for effective clinical management.”
Reviewer 2 Report (New Reviewer)
The aim of the study is clearly described and emphasized. Nevertheless, the authors could better provide a full definition of CH, its clinical features, and why CH is clinically relevant for hematological and cancer patients. Moreover, the role of CHIP in CVD and atherosclerosis should be better explained in the introduction.
The topic, the research question, and the study's objective are well explained but should be better justified. Although the references are correctly cited, appropriate and relevant, some pivotal studies necessary to fully comprehend the background are missing.
Please check:
1. Jaiswal, S.; Ebert, B.L. Clonal hematopoiesis in human aging and disease. Science 2019.
2. Steensma, D.P. Clinical consequences of clonal hematopoiesis of indeterminate potential. Blood Adv. 2018, 2, 3404–3410.
A more recent literary review on the topic might be:
3. Papa V, Marracino L et al. Translating Evidence from Clonal Hematopoiesis to Cardiovascular Disease: A Systematic Review. J Clin Med. 2020 Aug; 9(8): 2480.
The methodology is well designed and described; results are well presented. Nevertheless, the text (especially the supplemented material) is written in a conversational language and would benefit from extensive editing by a native English-speaking editor. Moreover, it should be written in a scientific style, using a passive form and the 3rdperson. Please avoid personal pronouns such as “we,” “our,” etc.
Author Response
- The aim of the study is clearly described and emphasized. Nevertheless, the authors could better provide a full definition of CH, its clinical features, and why CH is clinically relevant for hematological and cancer patients. Moreover, the role of CHIP in CVD and atherosclerosis should be better explained in the introduction.
Thank you for your evaluation and your helpful comments. We added some phrases in the introduction (line 41, 43) to address this comment.
- The topic, the research question, and the study's objective are well explained but should be better justified. Although the references are correctly cited, appropriate and relevant, some pivotal studies necessary to fully comprehend the background are missing.
Thank you for these interesting citations. We have added an additional conclusion in the discussion (line 261-265) and referred to the manuscripts by Steensma and Papa.
- The methodology is well designed and described; results are well presented. Nevertheless, the text (especially the supplemented material) is written in a conversational language and would benefit from extensive editing by a native English-speaking editor. Moreover, it should be written in a scientific style, using a passive form and the 3rdperson. Please avoid personal pronouns such as “we,” “our,” etc.
The supplement was extensively revised in accordance with the suggestion. The changes were not marked due to the extensive changes. No additional information was added.
Reviewer 3 Report (New Reviewer)
Manuscript by Büttner et al. “Clonal Hematopoiesis Mutations Are Present in Atherosclerotic Lesions in Peripheral Artery Disease” is an interesting study concerning the occurrence of mutations common for clonal hematopoiesis in patients diagnosed and treated for peripheral artery disease.
For patients demonstrating CH mutations in PB, the same CH mutations present in atherosclerotic plaques were observed in the majority of cases. The observation of CH mutations in atherosclerotic lesions is expected, unless we assume that these mutations subvert the myeloid cell physiology to such an extent that these cells are no longer able to infiltrate the lesions, which is apparently not the case.
However, there is a related and quite important issue as to whether the CH aggravates atherosclerosis directly through participation of myeloid cells with CH mutations in lesion progression, or indirectly through systemic elevation of pro-inflammatory milieu. If the first alternative were true, one would expect to find the higher VAF in myeloid cells in plaques compared to PB. Unfortunately, the results reported in the manuscript do not provide evidence for either alternative. The VAF in plaques, as follows from Table 2, is often lower than in PB, but the reported values for plaques have to be corrected according to the real contribution of myeloid cells to the plaque material. However, this analysis was not performed and the relevant data are lacking.
Another issue is a prevalence of CH mutations in patients with peripheral artery disease. According to findings reported in this work, 45% of these patients have CH mutations in PB, which appears to be pretty high. However, a matching cohort of healthy individuals have not been analyzed using the same methodology in this study, while the results in the literature seem to vary significantly for similarly aged humans, ranging from 6-12% to about 50% individuals harboring CH mutations, dependent on the sensitivity of analysis. Thus, correct interpretation of reported data depends on the sensitivity of the approaches used and does not seem to be straightforward. Authors should at least elaborate in more detail on comparison of their results to literature data.
Given the limitations of this study as discussed by the authors themselves, the more tentative title might deserve consideration, with terms like “A pilot study” or “Preliminary data”.
Minor remarks
According to the Table 2, for a number of patients, the VAF was higher in enriched cell populations than in peripheral blood. Thus, mathematically, the VAF in the depleted populations (Residuals) should have been lower than in PB. However, we usually observe the opposite pattern, which is rather strange. Authors seem to discuss this (rather indirectly) in lines 170-177 of the text, however, this part looks rather ambiguous and not convincing enough; more clear-cut discussion of this issue is warranted.
Titles in the main text & Supplementary are not identical.
In my version of manuscript, many places in text are highlighted yellow or green. Does this mean anything?
Author Response
- Manuscript by Büttner et al. “Clonal Hematopoiesis Mutations Are Present in Atherosclerotic Lesions in Peripheral Artery Disease” is an interesting study concerning the occurrence of mutations common for clonal hematopoiesis in patients diagnosed and treated for peripheral artery disease.
Thank you for your positive evaluation and your comments.
- For patients demonstrating CH mutations in PB, the same CH mutations present in atherosclerotic plaques were observed in the majority of cases. The observation of CH mutations in atherosclerotic lesions is expected, unless we assume that these mutations subvert the myeloid cell physiology to such an extent that these cells are no longer able to infiltrate the lesions, which is apparently not the case. However, there is a related and quite important issue as to whether the CH aggravates atherosclerosis directly through participation of myeloid cells with CH mutations in lesion progression, or indirectly through systemic elevation of pro-inflammatory milieu. If the first alternative were true, one would expect to find the higher VAF in myeloid cells in plaques compared to PB. Unfortunately, the results reported in the manuscript do not provide evidence for either alternative. The VAF in plaques, as follows from Table 2, is often lower than in PB, but the reported values for plaques have to be corrected according to the real contribution of myeloid cells to the plaque material. However, this analysis was not performed and the relevant data are lacking.
The reviewer has addressed an important ongoing discussion. We are aware that our study is descriptive in nature and cannot elucidate cellular pathomechanisms. This was discussed (see lines 227, 239) and the shortcomings of the study are discussed as limitation (line 258). Nevertheless, we would like to point out that the presence of CH mutations in atherosclerotic lesions, although likely, has only been demonstrated in human material with the present study.
- Another issue is a prevalence of CH mutations in patients with peripheral artery disease. According to findings reported in this work, 45% of these patients have CH mutations in PB, which appears to be pretty high. However, a matching cohort of healthy individuals have not been analyzed using the same methodology in this study, while the results in the literature seem to vary significantly for similarly aged humans, ranging from 6-12% to about 50% individuals harboring CH mutations, dependent on the sensitivity of analysis. Thus, correct interpretation of reported data depends on the sensitivity of the approaches used and does not seem to be straightforward. Authors should at least elaborate in more detail on comparison of their results to literature data.
A comparison with a healthy control cohort would indeed be helpful to interpret the results of this study, but is difficult to realize for several reasons. The focus of the study was on atherosclerotic lesions in patients with PAD who underwent surgery. Surgical procedures in which sampling of non-symptomatic vessels or non-symptomatic atherosclerotic lesions is ethically justifiable may be conceivable but were beyond the scope of this study. Nevertheless, the question should be addressed in future studies.
In fact, the detection rate for CH mutations in peripheral blood varies greatly depending on the detection method used. We have discussed this (line 144-149) and attempted to overcome this obstacle by comparing results in patients with PAD with those made in patients with cardiogenic shock complicating acute myocardial infarction using the same technical approach (line 150-157). Both cohorts depict cardiovascular disease based on atherosclerotic alterations while age and comorbidities are comparable. At least with respect to this cohort, which in many respects shows a very high degree of similarity, comparisons and conclusions are permissible.
- Given the limitations of this study as discussed by the authors themselves, the more tentative title might deserve consideration, with terms like “A pilot study” or “Preliminary data”.
We discussed this suggestion but decided against because the pilot nature is mentioned at several points in the manuscript.
Minor remarks
- According to the Table 2, for a number of patients, the VAF was higher in enriched cell populations than in peripheral blood. Thus, mathematically, the VAF in the depleted populations (Residuals) should have been lower than in PB. However, we usually observe the opposite pattern, which is rather strange. Authors seem to discuss this (rather indirectly) in lines 170-177 of the text, however, this part looks rather ambiguous and not convincing enough; more clear-cut discussion of this issue is warranted.
Thank you for this important comment. As we stated in the mentioned part of the discussion an extrapolation of VAF in assorted cell populations to peripheral blood is not possible and at best semi-quantitative. It is unclear to what extent specific cell-subpopulations are lost during the enrichment process. Furthermore, the VAF determination may also be subject to fluctuations, especially near the detection limit. We cannot describe these influencing factors precisely, which is why we have refrained from an extensive discussion.
- Titles in the main text & Supplementary are not identical.
Thank you for the comment. The title was changed in the supplement.
- In my version of manuscript, many places in text are highlighted yellow or green. Does this mean anything?
During the first revision process, we took the opportunity to remove typos and reword some sentences to improve readability. These changes were highlighted in yellow. The changes that were made based on reviewer comments were highlighted in green.
Round 2
Reviewer 2 Report (New Reviewer)
Suggestions have been properly assessed.
Therefore, I suggest accepting the revised manuscript in its present form.
This manuscript is a resubmission of an earlier submission. The following is a list of the peer review reports and author responses from that submission.
Round 1
Reviewer 1 Report
In this paper, the authors examined the presence of CHIP mutations in blood samples and biopsies from patients with PAD undergoing surgical revascularization. Based on sequencing of the total DNA extracted from blood cells and tissues, authors show that CHIP mutations in the blood cells are frequently also present in the atherosclerotic lesion and surrounding tissues. Establishing whether immune-derived cells carrying CHIP mutations are present in atherosclerotic plaque and, ultimately, whether they play a role in PAD is important but technically challenging goal.
This study has major limitations and methodological weaknesses that make interpreting the results difficult. Therefore, in the present form, the study still seems preliminary.
My main concern is that the cell composition of blood subpopulation enrichments and tissues is unknown. In addition, in a large fraction of patients, mutations are present at similar abundancy in atherosclerotic plaque, subcutaneous tissue and perivascular fat. This makes challenging to link PAD to the presence of the mutations.
My comments on specific aspects that should be addressed before the paper can be considered for publication:
-How many immune-derived cells, particularly macrophages, are present in the different samples? The composition of the tissues should be characterized (for instance, by immunohistochemistry or flow cytometry). This quantification would allow estimating whether immune cells with CHIP mutations are present in the examined tissues more frequently than wild-type cells.
-What is the fraction of CD14, CD34 or "residual" cells in total blood cells? The composition of blood subpopulations should be defined for each sample by flow cytometry.
-Some results are difficult to explain. For instance, TET2 mutation in ID11 is detectable only in "residuals" cells but not in the whole blood, while TET2 mutation in ID9 is only detectable in CD34 cells but not in whole blood. In ID 28, VAF % of DNMT3A mutations is higher in whole blood compared to the 3 enrichments (If I'm not missing something, the mutation should be present with a higher VAF% than PB in at least one of the 3 enrichment).
-The presence of blood cells in different tissues (possibly to a different extent from sample to sample) could alter the quantification. Has this aspect been considered?
Reviewer 2 Report
In general, the manuscript is readable. However It Is missing of several contents. The introduction Is really sloppy and must be revised. The references are insufficient. The sample size Is top small and results are not conclusive. The discussion Is missing of fundamentals.